# An all 2D bio-inspired gustatory circuit for mimicking physiology and psychology of feeding behavior

Subir Ghosh [1], Andrew Pannone[1], Dipanjan Sen[1], Akshay Wali [2], Harikrishnan Ravichandran[1] & Saptarshi Das [1,2,3,4] ✉

Animal behavior involves complex interactions between physiology and psychology. However, most AI systems neglect psychological factors in decision-making due to a limited understanding of the physiological-psychological connection at the neuronal level. Recent advancements in brain imaging and genetics have uncovered specific neural circuits that regulate behaviors like feeding. By developing neuro-mimetic circuits that incorporate both physiology and psychology, a new emotional-AI paradigm can be established that bridges the gap between humans and machines. This study presents a bio-inspired gustatory circuit that mimics adaptive feeding behavior in humans, considering both physiological states (hunger) and psychological states (appetite). Graphene-based chemitransistors serve as artificial gustatory taste receptors, forming an electronic tongue, while $1L\text{-}MoS_2$ memtransistors construct an electronic-gustatory-cortex comprising a hunger neuron, appetite neuron, and feeding circuit. This work proposes a novel paradigm for emotional neuromorphic systems with broad implications for human health. The concept of gustatory emotional intelligence can extend to other sensory systems, benefiting future humanoid AI.

Physiology and psychology both play equally influential roles in human behavior and decision-making. Physiology is related to the physical states of the body such as wakefulness, hunger, body temperature, etc., which are measurable entities, whereas psychology is related to the emotional states of the mind such as satiety, happiness, fear, etc., which are mostly immeasurable. Yet, in the brain, sensory information processing and decision-making are controlled by both, indicating that there exists a physical connection between physiology and psychology. The current neurological understanding is that physiology is the driving force behind neural activities, whereas psychology plays a regulatory role through, for example, the increase or decrease of specific inhibitory or excitatory neurotransmitter release at the chemical synapses[1,2]. Despite the tight coupling between physiology and psychology, the current paradigms of artificial intelligence (AI), neuromorphic computing, and bio-inspired devices largely discard psychological factors in their design. As a result, much effort in AI is invested towards massive data-driven learning for accurate inference or decision-making with limited scope to include emotional intelligence[3]. Developing neuro-mimetic circuits that can integrate the influence of both physiology and psychology for sensory information processing and decision-making can, therefore, establish a new emotional-AI paradigm that can bridge the gap between humans and machines.

The first step towards the incorporation of emotional intelligence in future AI systems is to understand and identify the cortical connections between physiological and psychological factors that govern human behavior in response to one or more sensory stimuli. Arguably, these connections are often abstract as human behavior is easy to

[1]Engineering Science and Mechanics, Penn State University, University Park, PA 16802, USA. [2]Electrical Engineering, Penn State University, University Park, PA 16802, USA. [3]Materials Science and Engineering, Penn State University, University Park, PA 16802, USA. [4]Materials Research Institute, Penn State University, University Park, PA 16802, USA. ✉e-mail: sud70@psu.edu

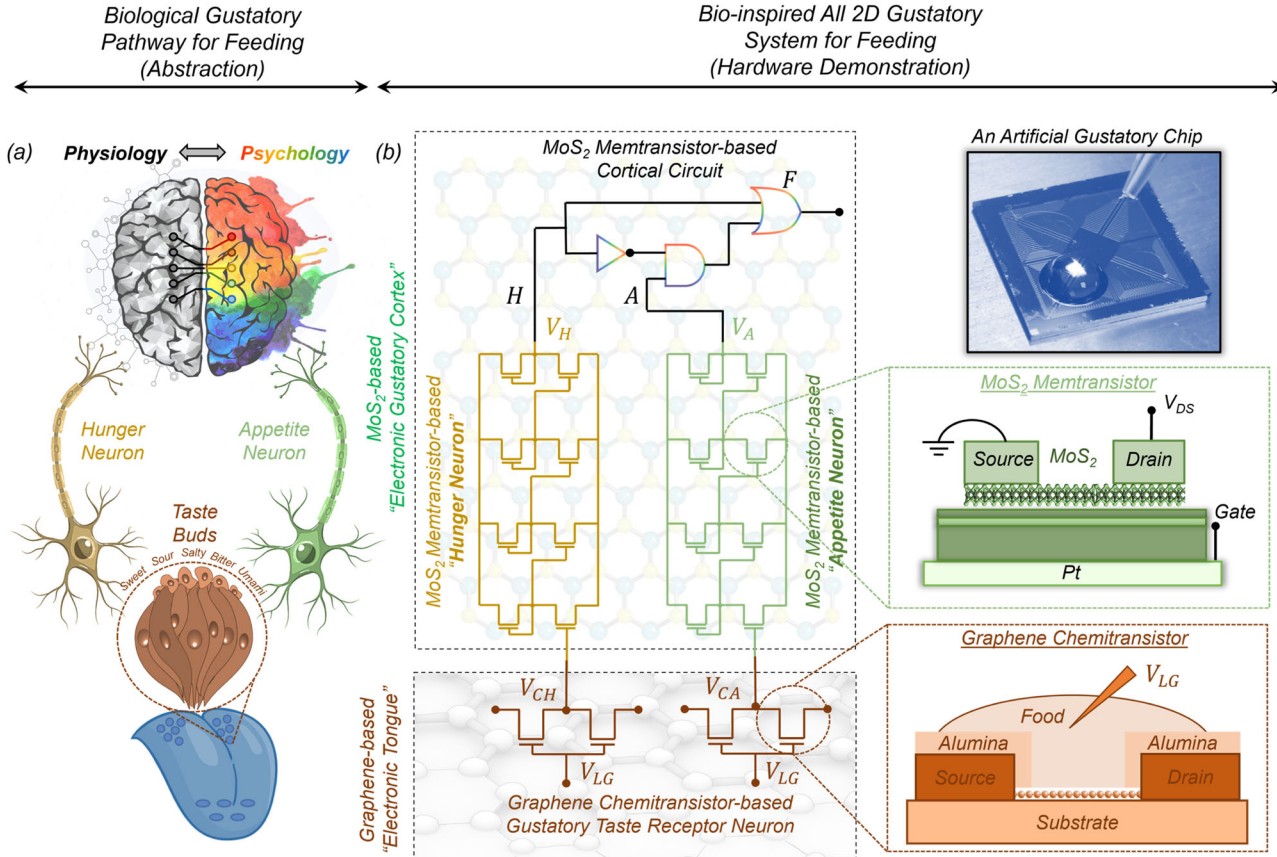

**Fig. 1 | Biological and bio-inspired gustatory systems for feeding. a** A simplified abstraction of the biological gustatory pathway for feeding. Taste receptors in the tongue generate electrical responses to food stimulants based on the chemical composition of the food, which is conveyed to both the hunger neuron and the appetite neuron, representing the physiological and psychological state of the body. The output from these neurons is subsequently evaluated by a cortical circuit, which makes the decision for feeding. **b** A bio-inspired gustatory system for making feeding decisions by exploiting heterogeneous integration of graphene chemitransistor-based "electronic tongue" for sensing and encoding taste stimulants and monolayer $MoS_2$ memtransistor-based "electronic gustatory cortex" for adaptive (in-memory) cortical computing. Additionally, the non-volatile and analog programmability of $MoS_2$ memtransistors allows for the adjustment of hunger and appetite thresholds, which is necessary to capture the adaptive feeding behavior.

observe and difficult to measure. However, it is possible to construct logic diagrams based on such observations to establish how and where psychological factors interject the sensory pathways to influence decision-making. Let us consider feeding as an example. Feeding is not only a complex human behavior that differs among cultures, communities, and countries, and is influenced by socioeconomic conditions, it is also controlled by physiological factors such as hunger and emotional factors such as preference or appetite[4–8]. Food that we like such as sweet, triggers feeding, whereas food that we dislike such as bitter, terminates further consumption of food. However, when hungry, our feeding behavior changes, and we accept food even if we dislike it. Therefore, if hunger ($H$) and appetite ($A$) are two state variables, representing the physiological and psychological factors respectively, then feeding ($F$) can be described as $F = H + \bar{H}A$, where $\bar{H}$ is the logical complement of $H$. In other words, if we are hungry, i.e., $H = 1$, then $F = 1$, i.e., feeding continues irrespective of our appetite, whereas, if we are not hungry, i.e., $H = 0$, then $F = 1$ only if $A = 1$, i.e., feeding happens if we have an appetite for the food. Finally, the volume of food that we intake or the duration of our feeding depends on how long $H$ and $A$ remain in their respective logical states, which in turn is determined by bodily thresholds that can differ widely across the population. Developing neural circuits that can mimic feeding behavior can, therefore, institute a new paradigm for emotional neuromorphic systems and at the same time have widespread consequences for human health. Note that the concept of gustatory emotional intelligence introduced in this work can also be translated to

other sensory systems including visual, audio, tactile, and olfactory emotional intelligence to aid future humanoid AI.

While the basic anatomy and physiology of the mammalian gustatory system are well understood, the precise neuronal circuits and molecular mechanisms that underlie hunger perception and appetite control remain the subject of intense investigation. However, there exist numerous well-documented psychological studies that highlight the logical connections between hunger and appetite[9–11]. Combining the two and using some of our own imagination, a simplified abstraction of the biological gustatory pathway for feeding is depicted in Fig. 1a. As shown in the figure, taste receptors in the tongue generate a response to food by encoding the chemical features of the food (e.g. sweet versus bitter) into corresponding electrical signals, which are conveyed to both the hunger neuron and the appetite neuron that represent the physiological and psychological state of the body, respectively. The output from these neurons is subsequently evaluated by a cortical circuit that makes the decision for feeding. A corresponding neuro-mimetic gustatory system based on the heterogenous integration of graphene chemitransistors for sensing and monolayer $MoS_2$ memtransistors for adaptive (in-memory) cortical computing is shown in Fig. 1b. Note that, graphene chemitransistors serve as artificial gustatory taste receptors, whereas the integrated circuits made using monolayer $MoS_2$ memtransistors serve as the hunger neuron and the appetite neuron and also emulate the cortical decision circuit for feeding. In other words, graphene chemitransistors are used as an electronic tongue, whereas $MoS_2$ memtransistors are used as an

electronic gustatory cortex. Additionally, the non-volatile and analog programmability of $MoS_2$ memtransistors allows for the adjustment of hunger and appetite thresholds, which is necessary to capture the adaptive behavior of any individual as well as feeding diversity across the population.

The choice of graphene, a single layer of hexagonally packed carbon atoms, as the electronic tongue is motivated by its remarkable bio/chemical sensing properties[12], which can be attributed to its (1) exceptionally high surface-to-volume ratio[13], (2) high carrier mobility at room temperature[14], and (3) inherently low electrical noise[15]. However, graphene lacks a bandgap, thus developing integrated circuits based on graphene to perform computation is challenging. In contrast, two-dimensional (2D) transition metal dichalcogenides (TMDs) such as $MoS_2$ offer a finite bandgap and excellent semiconducting properties, which are essential for the development of computing devices[16]. In recent years, such $MoS_2$-based devices have enabled various neuromorphic and bio-inspired applications through the integration of sensing, compute, and storage capabilities[17–23]. Additionally, $MoS_2$ is among the most mature two-dimensional (2D) materials, which can be grown at wafer scale using chemical vapor deposition techniques[24] and at the same time $MoS_2$-based aggressively scaled transistors[25] with near Ohmic contacts[26] have achieved high performance that meets the IRDS standards for advanced technology nodes[27–29]. It is also worth mentioning some recent studies based on flexible organic electronic systems that imitate sensory nerves by gathering pressure data and transforming it into action potentials to facilitate intricate tactile processing[30] and create a neuromorphic gustatory system that employs gel sensors and also in the literature there are artificial synapses to identify salt taste, analyze data, and issue alerts regarding excessive consumption[31].

## Results

### Graphene chemitransistors as artificial gustatory taste receptors

Taste receptors in the tongue serve as the primary interface responsible for interaction between the human gustatory system and the ingested food species during feeding. Chemical information collected by these sensory cells is translated into an electrical signal and transmitted via afferent neurons to the gustatory cortex where cortical circuits facilitate the perception of taste. Diverse interactions between taste receptors and chemicals found in foods enable the interpretation of distinctive tastes that are categorized broadly into five basic classes: sweet, salty, sour, bitter, and umami. To mimic the biological tongue, we exploit the remarkable chemisensing properties of graphene and create an artificial gustatory taste receptor comprising two graphene chemitransistors connected in series as shown using the optical image and circuit diagram in Fig. 2a, b, respectively (see the "Methods" section for additional details regarding the fabrication of graphene chemitransistors). Both the channel length ($L_{CH}$) and width ($W_{CH}$) for each graphene chemitransistor were designed to be 20 μm. Supplementary Information 1 shows the Raman spectrum obtained using a 532 nm wavelength laser excitation from the channel area of the graphene chemitransistor. The presence of strong peaks at ~1583 and ~2674 cm$^{-1}$ that correspond to the G-band and 2D-band, respectively, are characteristic features of $sp^2$ hybridized carbon atoms and confirm the presence of monolayer graphene in the channel of the chemitransistor[32].

For the feeding experiments, the graphene-based gustatory taste receptor is presented with an aqueous solution containing specific food species, which are diluted with de-ionized (DI) water. Dilution of food with DI water follows a similar process to biological gustation, which is facilitated by the mixture of foods with saliva. Interestingly, such chemical solutions can serve as liquid gates for graphene chemitransistors. An electrical bias applied through the liquid solution promotes the formation of an electric double layer (EDL) at the interface between the monolayer graphene channel and the liquid. This EDL acts as an ultra-thin gate dielectric allowing for the electrostatic control of the channel conductance. Figure 2c shows the transfer characteristics, i.e., the source-to-drain current ($I_{DS}$) as a function of the gate voltage applied to the liquid solution ($V_{LG}$) for a constant source-to-drain voltage ($V_{DS}$) of 500 mV for the two graphene chemitransistors that constitute the artificial gustatory taste receptor in the presence of DI water. The liquid gate voltage range for the operation of the artificial taste receptor was carefully selected to ensure that the gate leakage current remains significantly smaller than $I_{DS}$ for all taste stimulants as we will elucidate later. The slight variation in the transfer characteristics of the two graphene chemitransistors can be attributed to many factors such as metal impurity doping introduced by the copper etchant chemistry or polymer residues and defects introduced during the transfer and fabrication processes[33–39]. While device-to-device variation is undesired for many applications, it benefits the operation of our artificial taste receptors as we will explain next.

Figure 2d shows the response curve for the artificial taste receptor, i.e., the output voltage ($V_C$) measured at the common node terminal as a function of $V_{LG}$ with a supply voltage of 500 mV. The unusual shape of this response curve is a direct consequence of the device-to-device variation, in the absence of which the output from the artificial taste receptor will transform into a horizontal straight line with $V_C$ = 250 mV. Figure 2e–g, respectively, show the transfer characteristics of individual graphene chemitransistors and the response curve of the artificial taste receptor for all five taste categories, i.e., sweet, salty, sour, bitter, and umami (see Supplementary Information 2 for the corresponding gate leakage current for all taste categories which show negligible values in the range of ~1–10 nA within the $V_{LG}$ range). We have used diluted aqueous solutions of sucrose for sweet, coffee for bitter, lemon juice for sour, NaCl for salty, and soy sauce for umami. Note that after each experiment with the respective taste stimulants, the graphene chemitransistors undergo a rinsing process, including a 5-min rinse with deionized water (DI), followed by a 5-min treatment with acetone and a 3-min treatment with isopropyl alcohol (IPA). Supplementary Information 3 shows optical images of the sample taken before and after the experiment and after cleaning. Clearly, the washing procedure successfully removes any remaining solution precipitates, ensuring that subsequent testing remains unaffected. This is further highlighted in Supplementary Information 4, where the reusability of a representative graphene chemitransistor is assessed for different solution species over multiple cycles. A single cycle constitutes measuring the transfer characteristics using four different solution species. The electrical characterization was performed for a total of four cycles, showing that the chemitransistor characteristics for any given species remain mostly unaltered between cycles. Finally, the life cycle of the representative chemitransistor was evaluated through an endurance measurement, where it was measured for 100 cycles with a sugar solution as shown in Supplementary Information 5.

A physics-based semi-empirical model was also developed to capture the transfer characteristics of individual liquid-gated graphene chemistransistors and the response curve of the artificial taste receptor as described in Supplementary Information 6[40]. Supplementary Information 7 shows the impact of variation in several device-related fitting parameters for the two graphene chemitransistors on the response curve of a graphene-based artificial taste receptor. Some of these parameters include Dirac voltage ($V_{Dirac}$), which is defined as the liquid gate voltage ($V_{LG}$) corresponding to the minimum graphene channel conductance, and electron ($\mu_N$) and hole ($\mu_P$) carrier mobility values. Clearly, variation in $V_{Dirac}$ between the two graphene chemitransistors is the key towards obtaining a nonmonotonic response curve from the taste receptor, which flattens for $\Delta V_{Dirac} = 0$ V. Nevertheless, the individual graphene chemitransistors serve as the fundamental sensory elements for taste differentiation, whereas the taste receptor circuit is employed to generate corresponding unique $V_C$ for

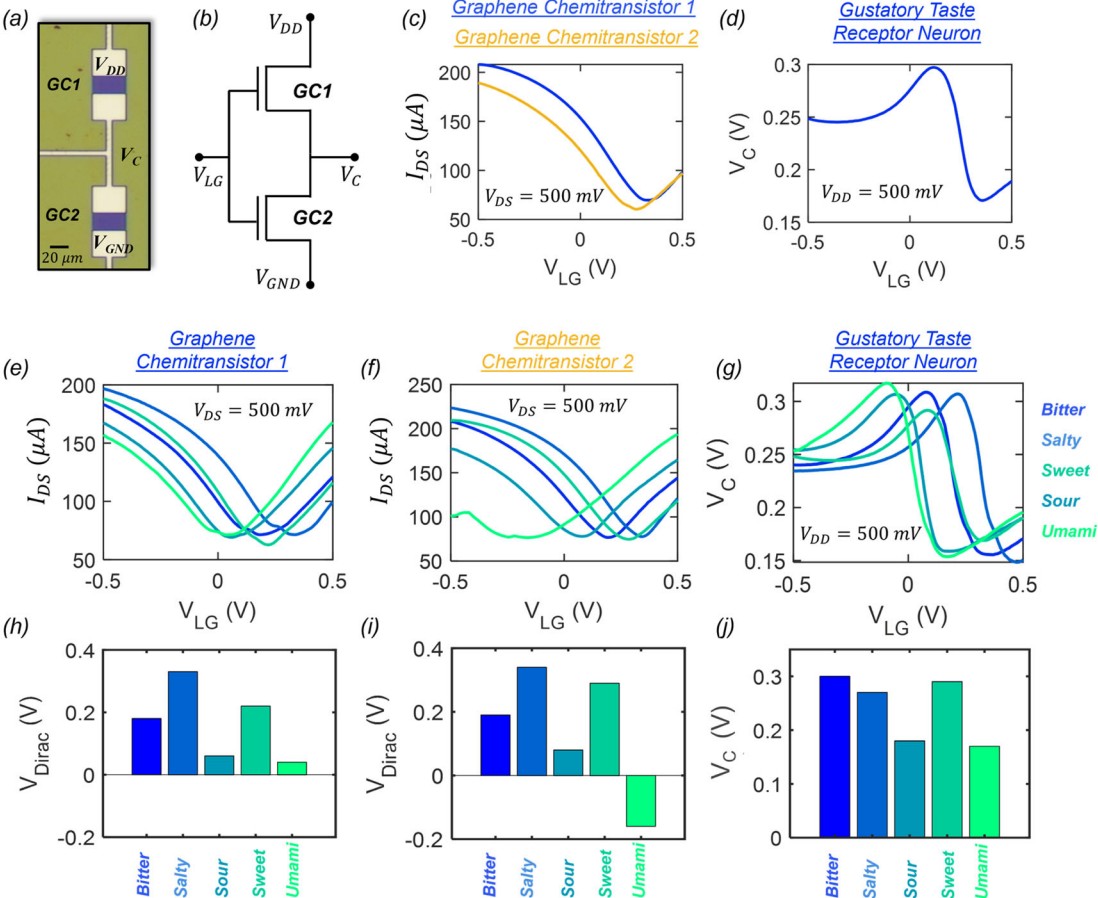

**Fig. 2 | Graphene chemitransistor-based "electronic tongue" for taste differentiation. a** Optical image and **b** circuit schematic of an artificial gustatory taste receptor composed of two graphene chemitransistors connected in series. **c** Transfer characteristics, i.e., source-to-drain current ($I_{DS}$) plotted as a function of the gate voltage applied to the liquid solution ($V_{LG}$) for a constant source-to-drain voltage ($V_{DS}$) of 500 mV for two graphene chemitransistors with DI water as the gating liquid. **d** Response curve for the artificial taste receptor where the output voltage ($V_C$) is measured as a function of $V_{LG}$ for a $V_{DD}$ of 500 mV. Transfer characteristics of **e** graphene chemitransistor 1 and **f** graphene chemitransistor 2 and **g** the response curve for the artificial taste receptor subjected to five taste stimulants i.e., sweet, salty, sour, bitter, and umami. The corresponding $V_{Dirac}$ values extracted from the transfet characteristics of **h** graphene chemitransistors 1 and **i** graphene chemitansistor 2 and **j** $V_C$ obtained at $V_{LG} = 0.1$ V. confirming the distinctness of each taste stimuli. The nonoverlapping distributions of $V_{Dirac}$ for each chemitransistor and unique $V_C$ values obtained from the artificial taste receptor confirm successful taste differentiation and validate the use of graphene chemitransistors as the "electronic tongue" for our demonstration.

further processing by the cortical circuits constructed using $MoS_2$ memtransistors for hunger and appetite perception and feeding decision. Figure 2h–j, respectively show the $V_{Dirac}$ values extracted for each of the two graphene chemitransistors from Fig. 2e, f and the $V_C$ obtained at $V_{LG} = 0.1$ V from Fig. 2g. The nonoverlapping distributions of $V_{Dirac}$ for each chemitransistor confirm that each taste stimulus has a distinct interaction with the graphene channel, obviating the need for surface functionalization. Furthermore, the $V_C$ generated by the artificial taste receptor are unique for each taste, which validates the use of graphene chemitransistors as the electronic tongue for our gustatory neural circuit for feeding.

Next, to invoke the influence of physiology and psychology on taste perception, we introduce two distinct pairs of artificial taste receptors that project onto hunger neurons and appetite neurons, respectively, as shown in Fig. 1b. At the same time, to simplify our demonstration, we confine our study to two taste perceptions, namely sweet and bitter. Finally, the feeding process is imitated through the temporal evolution of taste perception by exploiting the gradual evaporation of the liquid solution placed on top of the graphene-based artificial taste receptors. Figure 3a, b, respectively, show the temporal

changes in the response curve of the artificial taste receptor that projects onto the hunger neuron when subjected to sweet and bitter taste stimulants. The continuous shift in the response curves can be attributed to the changing concentration of the chemical species within the solution as the water evaporates, which leads to a change in the surface charge transfer doping of the graphene channel and hence the $V_{Dirac}$. Once the evaporation is complete, the gating is completely lost and the response curve becomes a flat line as a function of $V_{LG}$.

Figure 3c, d, respectively, show the corresponding temporal evolution of the output voltage, $V_{CH}$, measured at $V_{LG} = 0$ V for sweet and bitter taste stimulants. In both cases, the output, $V_{CH}$, shows a slow decrease before dropping abruptly to ~180 mV once the stimulant evaporates. The output of this taste receptor is fed as the input to the hunger neuron, which is a comparator circuit designed using monolayer $MoS_2$-based memtransistors as we will discuss later. The "hunger neuron" is characterized by a hunger threshold, $V_{HN}$, that determines its logical output state, $H$. For $V_{CH} > V_{HN}$, $H = 1$, i.e., the subject remains hungry, whereas, for $V_{CH} < V_{HN}$, $H = 0$, i.e., the subject becomes satisfied. In other words, the "hunger neuron" controls the physiological state associated with feeding.

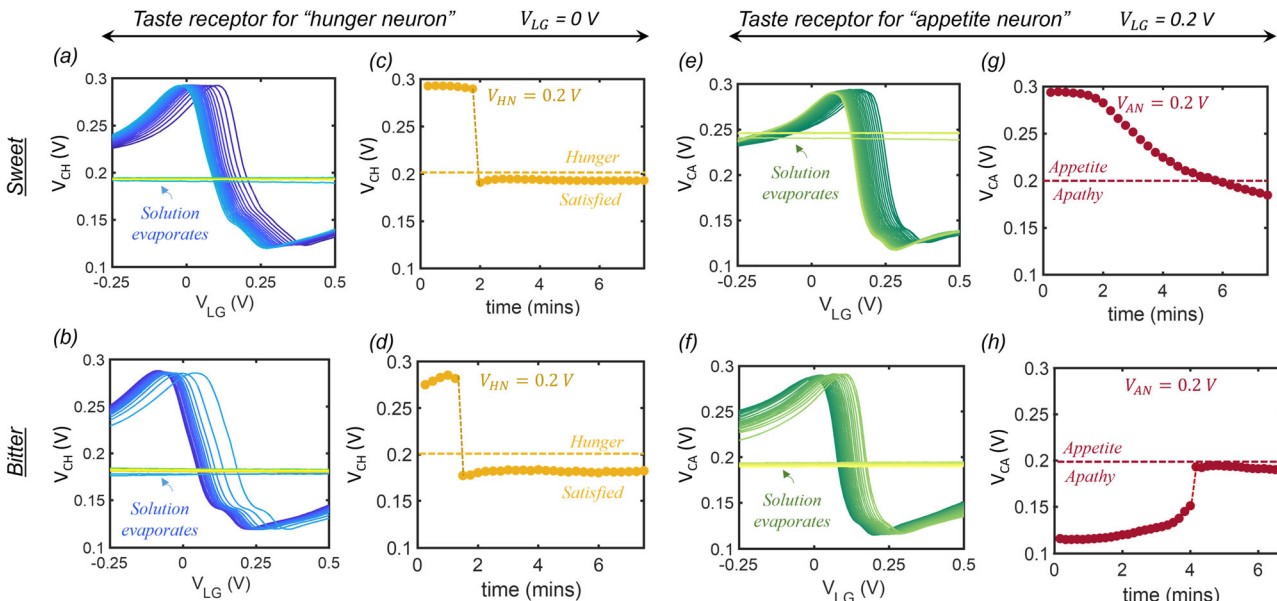

**Fig. 3 | Artificial taste receptors for "hunger neuron" and "appetite neuron".** Temporal changes in the response curve of the artificial taste receptor that projects onto the hunger neuron for **a** sweet and **b** bitter taste stimulants. The steady shift in the response curves is attributed to the change in the concentration of the chemical species within the solution which gradually evaporates leading to a change in the surface charge transfer doping of the graphene channel and hence the $V_{Dirac}$. Once the solution completely evaporates, there is a complete loss of any gating. Corresponding temporal evolution of output voltage ($V_{CH}$) measured at $V_{LG} = 0$ V for **c** sweet and **d** bitter taste stimulants where $V_{CH}$ shows a slow decrease before dropping abruptly to -180 mV for both the cases. Temporal changes in the response curve of the artificial taste receptor that projects onto the appetite neuron for **e** sweet and **f** bitter solutions showing similar steady shifts. Corresponding temporal evolution of output voltage ($V_{CA}$) measured at $V_{LG} = 0.2$ V for **g** sweet and **h** bitter taste stimulants showing the opposite monotonic trend.

In a similar way, we introduce the appetite neuron that receives its input from the second taste receptor as mentioned above, however, with a small but critical distinction in the magnitude of $V_{LG}$ used to generate the output voltage, $V_{CA}$. Figure 3e, f, respectively, show the temporal changes in the response curve of the artificial taste receptor that projects onto the appetite neuron for sweet and bitter taste stimulants. A similar shift is observed in the response curve of this taste receptor as the stimulants evaporate. However, for generating the output from this taste receptor, a different $V_{LG} = 0.2$ V is used. Fig. 3g, h, respectively, show the temporal evolution of $V_{CA}$ measured at $V_{LG} = 0.2$ V for sweet and bitter taste stimulants. Unlike the previous case, $V_{CA}$ shows a monotonic decrease from 300 to 180 mV for the sweet stimulant, whereas for the bitter stimulant, it shows a monotonic increase from 120 to 190 mV. In other words, the taste receptors are biased in such a manner that their response to similar taste stimulants generates dissimilar responses to differentiate between physiology and psychology. As we will discuss next, the output of this taste receptor is fed as the input to the appetite neuron, which is also a comparator circuit characterized by an appetite threshold, $V_{AN}$, that determines its logical output state, $A$. For $V_{CA} > V_{AN}$, $A = 1$, i.e., the subject feels appetite towards the taste, whereas, for $V_{CA} < V_{AN}$, $A = 0$, i.e., the subject is apathetic to the taste. In other words, the appetite neuron controls the psychological state associated with feeding. As we will elucidate in the next section, $V_{AN}$ can be adjusted by exploiting the programming capability of $MoS_2$ memtransistors to reflect the adaptive behavior and feeding diversity across the population.

**Monolayer $MoS_2$ memtransistor as artificial gustatory cortex**
As mentioned in the previous section, the hunger neuron and the appetite neuron are comparator circuits built using monolayer $MoS_2$ memtransistors. Figure 4a, b, respectively, show the optical image and circuit schematic of the comparator composed of 8 $MoS_2$ memtransistors ($MT_1$–$MT_8$). All memtransistors have channel length ($L_{CH}$) and width ($W_{CH}$) of 1 and 5 μm, respectively. Note that the monolayer $MoS_2$

utilized in this study was grown using a metal-organic chemical vapor deposition (MOCVD) technique on an epitaxial sapphire substrate at 1000 °C and subsequently transferred onto the artificial gustatory chip using a wet transfer procedure for the fabrication of memtransistors and all necessary cortical circuits. Details on monolayer $MoS_2$ synthesis, film transfer, and fabrication of memtransistors can be found in the Methods section and in our previous works[24,41–46]. Electrical characterization, which includes transfer characteristics, analog programmability, and non-volatile retention of $MoS_2$ memtransistors are shown in Supplementary Information 8.

Note that, in the comparator circuit, the memtransistors, $MT_1$, $MT_3$, $MT_5$, and $MT_7$ serve as depletion loads as their gate and drain terminals are shorted making them operate in their saturation regimes. Consequently, each memtransistor pair, $MT_1 - MT_2$, $MT_3 - MT_4$, $MT_5 - MT_6$, and $MT_7 - MT_8$ serve as an inverter and the comparator circuit can be considered to be a 4-stage cascaded inverter. While a 2-stage cascaded inverter can also be used as a comparator, the 4-stage cascaded inverter allows us to achieve higher gain, which is critical for generating cleaner digital signals at the output of the hunger neuron and the appetite neuron. Figure 4c–f, respectively, shows the voltages, $V_{N4}$, $V_{N5}$, $V_{N6}$, and $V_{N7}$, measured at nodes, N4, N5, N6, and N7, i.e., at the output of each inverter stage, as a function of the input voltage, $V_{N2}$, applied to node N2, i.e., the gate terminal of $MT_2$. Insets of Fig. 4c–f also refer to the peak gain for the corresponding stages (see Supplementary Information 9 for the gain plots for each of the four inverter stages). As expected, the peak gain increases monotonically from -14 in the first stage, to -29 in the second stage, -81 in the third stage, and -90 in the fourth stage. The increasing gain transmutes the gradual state transition observed at $V_{N4}$ into an abrupt state transition at $V_{N7}$ from 0 to $V_{DD}$, i.e., logic "0" to logic "1". The input voltage, $V_{N2}$, at which this sharp state transition occurs for $V_{N7}$ is referred to as the comparator threshold ($V_{TH-C}$). For the hunger neuron and the appetite neuron, $V_{TH-C}$ is represented by $V_{HN}$ and $V_{AN}$, respectively, and $V_{N7}$ is represented by logic state variables H and A, respectively.

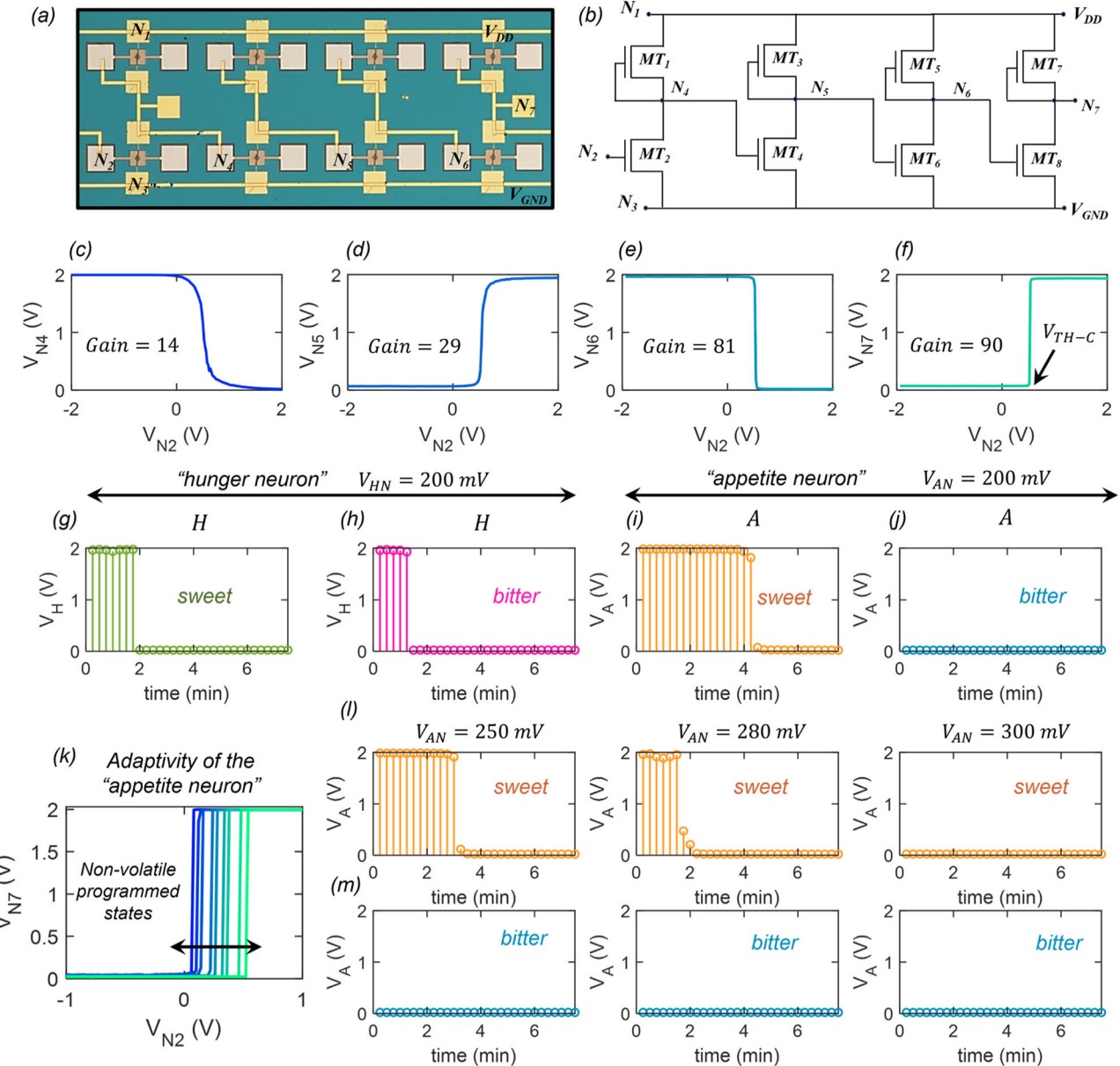

**Fig. 4 | Monolayer MoS$_2$ memtransistor-based "hunger neuron" and "appetite neuron". a** Optical image and **b** circuit schematic of a representative comparator circuit composed of 8 MoS$_2$ memtransistors (MT$_1$–MT$_8$) used as the "hunger neuron" and the "appetite neuron". The comparator circuit is a 4-stage cascaded inverter as each memtransistor pair, MT$_1$ – MT$_2$, MT$_3$ – MT$_4$, MT$_5$ – MT$_6$, and MT$_7$ – MT$_8$ serve as an inverter. **c** V$_{N4}$ measured at node, N4. **d** V$_{N5}$ measured at node, N5. **e** V$_{N6}$ measured at node, N6, and **f** V$_{N7}$ measured at node, N7, as a function of the input voltage, V$_{N2}$, applied to node N2. Peak gain for each inverter stage is mentioned in the inset. Highest gain of ~90 is observed for the output stage, which allows for the abrupt state transition for V$_{N7}$ from 0 to V$_{DD}$, i.e., logic "0" to logic "1". The input voltage, V$_{N2}$, at which this sharp state transition occurs is referred to as the comparator threshold (V$_{TH–C}$). For the "hunger neuron" and the "appetite neuron", V$_{TH–C}$ is represented by V$_{HN}$ and V$_{AN}$, respectively, and V$_{N7}$ is represented by logic state variables H and A, respectively. Temporal evolution of V$_H$, i.e., logic state of H in response to **g** sweet and **h** bitter taste stimulants. Temporal evolution of V$_A$, i.e., logic state of A in response to **i** sweet and **j** bitter taste stimulants. **k** Transfer curve for the "appetite neuron" for different V$_{AN}$ obtained through non-volatile programming of MoS$_2$ memtransistor, MT$_2$. Corresponding temporal evolution of V$_A$ or A in response to **l** sweet and **m** bitter taste stimulants.

Figure 4g, h show the temporal evolution of V$_H$, i.e., the logic state of the hunger neuron, H, in response to V$_{CH}$ obtained at the output of the artificial gustatory taste receptor (Fig. 3c, d) for sweet and bitter taste stimulants, respectively. The hunger threshold, V$_{HN}$, was set to 200 mV. Note that for both taste stimulants, H = 1 at the beginning since V$_{CH}$ > V$_{HN}$, which is indicative of the physiological state of hunger. However, after some time, ~2 min, V$_{CH}$ drops below V$_{HN}$, and H becomes 0, indicating satisfaction. Similarly, Fig. 4i, j show the temporal evolution of V$_A$, i.e., the logic state of the appetite neuron, A, in response to V$_{CA}$ obtained at the output of the other artificial gustatory taste receptor (Fig. 3g, h) for sweet and bitter taste stimulants, respectively. The appetite threshold, V$_{AN}$, was also set to 200 mV. For the bitter stimulant, V$_{CA}$ < V$_{AN}$ making A = 0 for the entire duration, which is indicative of psychological apathy towards that taste. In contrast, for the sweet stimulant, V$_{CA}$ > V$_{AN}$ resulting in A = 1 at the beginning of feeding indicating the presence of appetite. A becomes 0 only after sufficient consuming of sweet once V$_{CA}$ < V$_{AN}$. Also note that, for the sweet stimulant, between ~2–4 min, A = 1, even when H = 0, i.e., the appetite for sweet continues even when the hunger is satisfied. As we will see later, the psychological attribute of appetite ensures that the feeding continues beyond the physiological need dictated by hunger.

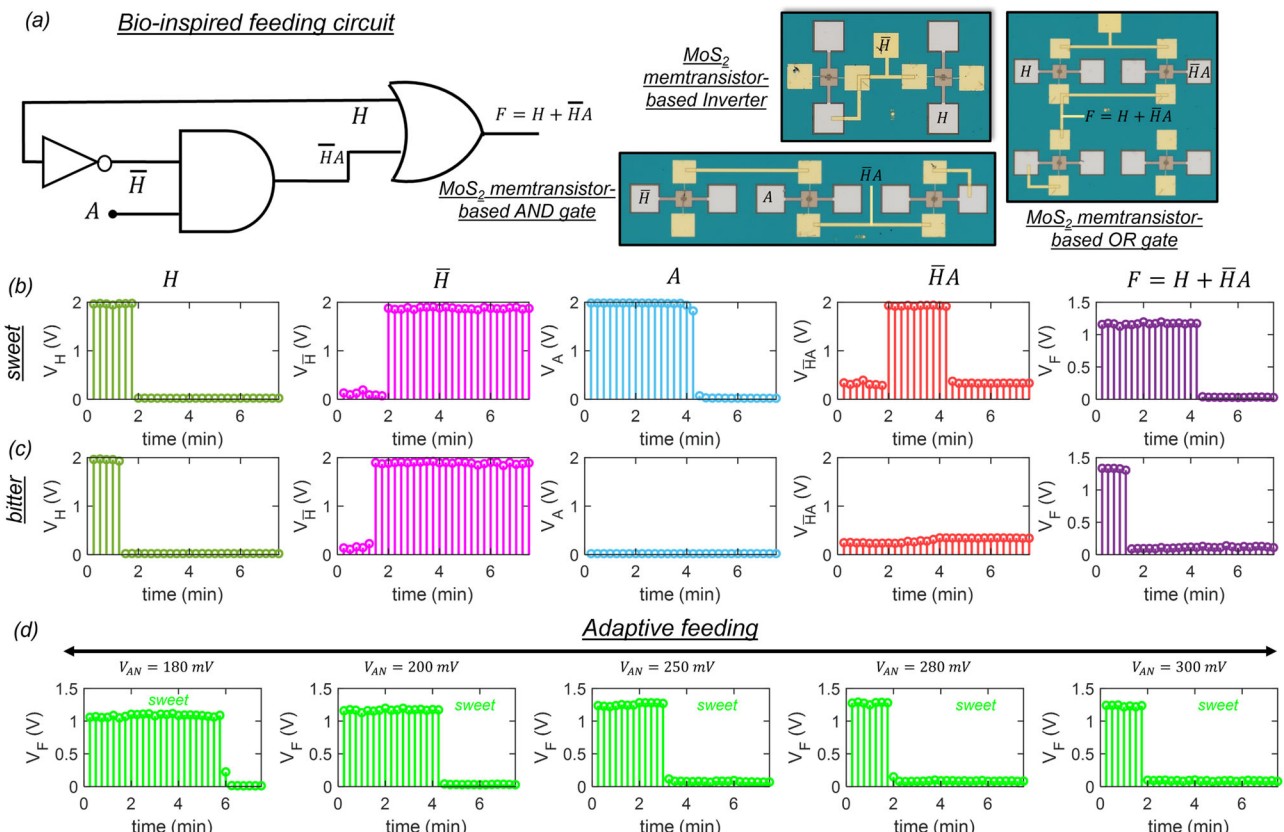

**Fig. 5 | Bio-inspired and adaptive feeding circuit. a** Schematic of the neuro-mimetic feeding circuit used for generating the feeding output, $F = H + \bar{H} A$, and optical images for the various circuit elements used to construct the feeding circuit, which include one inverter, one AND gate, and one OR gate. Temporal evolution of $H$, $\bar{H}$, $A$, $\bar{H} A$, and $F$, for **b** sweet and **c** bitter stimulants, respectively. Note that, for the bitter stimulant, the feeding terminates, i.e., $F = 0$ as soon as the hunger is satisfied, i.e., $H = 0$ due to apathy towards that taste. However, for sweet stimulant, the feeding continues, i.e., $F = 1$ even after the hunger is satisfied, i.e., $H = 0$. **d** Temporal evolution of $F$ for sweet corresponding to different appetite threshold, $V_{AN}$, mimicking adaptive feeding. Since a higher $V_{AN}$ shortens the appetite for sweet, the feeding terminates, i.e., $F = 0$, as soon as the hunger is satisfied, whereas for a lower $V_{AN}$ appetite for sweet is prolonged resulting in over-feeding beyond hunger.

Next, we introduce the aspect of adaptive feeding behavior by exploiting the non-volatile programmability of our monolayer $MoS_2$ memtransistors. Figure 4k shows the transfer curve for the appetite neuron for different $V_{AN}$ and Fig. 4l, m show the corresponding temporal evolution of $A$ in response to sweet and bitter taste stimulants, respectively. Note that, $A$ continues to remain at the logic state of 0 for the bitter stimulant, irrespective of $V_{AN}$, due to psychological apathy towards that taste. However, the time duration for which $A = 1$ for the sweet stimulant changes with $V_{AN}$. A lower $V_{AN}$ allows longer appetite for sweet, whereas a higher $V_{AN}$ shortens the appetite for sweet. In other words, $V_{AN}$ can be used as a knob to study the impact of psychological adaptation on feeding behavior.

### Bio-inspired and adaptive feeding circuit

Finally, we introduce the monolayer $MoS_2$-based integrated circuit that imitates feeding behavior based on the expression derived earlier, i.e., $F = H + \bar{H} A$. Figure 5a shows the schematic and optical images for the various circuit components used to construct the feeding circuit, which include one inverter, one AND gate, and one OR gate. Supplementary Information 10 demonstrates the operation of each of these circuit components. Figure 5b, c show the temporal evolution of $H$, $\bar{H}$, $A$, $\bar{H} A$, and $F$, for sweet and bitter stimulants, respectively. Note that, for the bitter stimulant, the feeding terminates, i.e., $F = 0$ as soon as the hunger is satisfied, i.e., $H = 0$ due to apathy towards that taste. However, for sweet stimulant, the feeding continues, i.e., $F = 1$ even after the hunger is satisfied, i.e., $H = 0$. The feeding terminates, i.e., $F = 0$ only when the appetite is over. In other words, feeding can occur without hunger if appetite is stimulated due to the presence of appealing foods.

Furthermore, Fig. 5d shows the temporal evolution of $F$ for sweet corresponding to different appetite threshold, $V_{AN}$. Since a higher $V_{AN}$ shortens the appetite for sweet, the feeding terminates, i.e., $F = 0$, as soon as the hunger is satisfied, whereas for a lower $V_{AN}$ appetite for sweet is prolonged resulting in over-feeding beyond hunger. Needless to say, our bio-inspired gustatory circuit for feeding can capture natural human behavior and serve as a platform to study human health. For example, an extreme appetite for sweets can lead to excessive feeding and may cause obesity over time. At the same time, diabetic individuals must resist their appetite towards sweets by behavioral changes.

Note that while we have used five basic tastes for our proof-of-concept demonstration, the concept is scalable to diverse range of taste stimulants. Supplementary Information 11 shows the transfer characteristics of a graphene chemitransistor for four different sugar solutions as well as four distinct milk (umami) solutions. Similarly, Supplementary Information 12 shows the transfer characteristics of a graphene chemitransistor for different concentrations of salt solutions. These results clearly highlight the scalability of our gustatory neuromorphic platform to more taste variants and concentrations. A more detailed and comprehensive analysis is planned as a future study. Finally, the concept of machines possessing emotional intelligence is an area of ongoing research and development[3,47–50]. While machines cannot experience emotions in the same way humans do, incorporating emotional intelligence into their capabilities can offer several potential benefits especially with growing dominance of artificial intelligence (AI) in all spheres of our lives (see Supplementary Information 13 for more discussion). Similarly, the rationale behind completely separating hunger and appetite as different state variables is discussed in Supplementary Information 14.

## Discussion

In conclusion, we have developed an artificial gustatory system for feeding that integrates the influence of both physiology and psychology for decision-making to bridge the gap between humans and machines. To the best of our knowledge, this is a first-of-its-kind demonstration that can enable a new emotional-AI paradigm and advance the development of humanoid robots. Our neuro-mimetic gustatory system is based on heterogeneous integration of monolayer graphene-based chemitransistors, which serve as an electronic tongue to differentiate between various tastes, and monolayer $MoS_2$-based memtransistors, which serve as an electronic gustatory cortex for performing analog and digital computations for making feeding decision. The concept of gustatory emotional intelligence introduced in this work can also be translated to other sensory systems including visual, audio, tactile, and olfactory emotional intelligence to aid future AI.

## Methods

### Fabrication of graphene chemitransistors

Using a wet transfer process, monolayer graphene was first transferred from a commercially procured copper foil onto the preferred p++ Si substrate with 285 nm of thermally grown $SiO_2$. Following that, the channel regions were patterned using electron beam (e-beam) lithography and oxygen plasma etching. Moreover, an additional step of e-beam lithography was also employed to define the source, drain, and gate electrodes, which was later followed by e-beam evaporation of a 40/30 nm Ni/Au metal stack. Subsequently, an alumina capping layer of 100 nm was deposited over the source and drain contacts as well as the peripheral leads, deciding to leave the gate electrode uncapped. A thick alumina capping layer has been used here to insulate the gate electrode from the source/drain contacts to minimize the gate leakage current in the presence of the liquid media.

### Fabrication of local back-gate islands for $MoS_2$ memtransistors

To define the back-gate island regions, the substrate (285 nm $SiO_2$ on $p^{++}$-Si) was spin-coated with a bilayer photoresist consisting of Lift-Off-Resist (LOR 5A) and Series Photoresist (SPR 3012) baked at 185 and 95 °C, respectively. The bilayer photoresist was then exposed using a Heidelburg Maskless Aligner (MLA 150) to define the island and developed using MF CD26 microposit, followed by a de-ionized (DI) water rinse. The back gate electrode of 20/50 nm Ti/Pt was deposited using e-beam evaporation. The photoresist was removed using acetone and Photo Resist Stripper (PRS 3000) and cleaned using 2-propanol (IPA) and DI water. An atomic layer deposition (ALD) process was then implemented to grow 40 nm $Al_2O_3$, 3 nm $HfO_2$, and 7 nm $Al_2O_3$ on the entire substrate including the island regions. To access the individual Pt back-gate electrodes, etch patterns were defined using the same bilayer photoresist consisting of LOR 5A and SPR 3012. The bilayer photoresist was then exposed using a Heidelburg MLA 150 and developed using MF CD26 microposit. 50 nm dielectric stack was subsequently dry-etched using the $BCl_3$ chemistry at 5 °C for 20 s, which was repeated four times to minimize heating in the substrate. Next, the photoresist was removed to give access to the individual Pt electrodes.

### Large area monolayer $MoS_2$ film growth

Monolayer $MoS_2$ was deposited on epi-ready 2" c-sapphire substrate by metalorganic chemical vapor deposition (MOCVD). An inductively heated graphite susceptor equipped with wafer rotation in a cold-wall horizontal reactor was used to achieve uniform monolayer deposition as previously described[51]. Molybdenum hexacarbonyl ($Mo(CO)_6$) and hydrogen sulfide ($H_2S$) were used as precursors. $Mo(CO)_6$ maintained at 10 °C and 950 Torr in a stainless-steel bubbler was used to deliver 0.036 sccm of the metal precursor for the growth, while 400 sccm of $H_2S$ was used for the process. $MoS_2$ deposition was carried out at 1000 °C and 50 Torr in $H_2$ ambient, where monolayer growth was achieved in 18 min The substrate was first heated to 1000 °C in $H_2$ and maintained for 10 min before the growth was initiated. After growth, the substrate was cooled in $H_2S$ to 300 °C to inhibit the decomposition of the $MoS_2$ films. More details can be found in our earlier work[19,24,52].

### $MoS_2$ film transfer to local back-gate islands

To fabricate the $MoS_2$ FETs, MOCVD-grown monolayer $MoS_2$ film was transferred from the sapphire to $SiO_2/p^{++}$-Si substrate with local back-gate islands using a polymethyl-methacrylate (PMMA)-assisted wet transfer process. First, $MoS_2$ on the sapphire substrate was spin-coated with PMMA and then baked at 180 °C for 90 s. The corners of the spin-coated film were scratched using a razor blade and immersed inside 1 M NaOH solution kept at 90 °C. Capillary action causes the NaOH to be drawn into the substrate/film interface, separating the PMMA/$MoS_2$ film from the sapphire substrate. The separated film was rinsed multiple times inside a water bath and finally transferred onto the $SiO_2/p^{++}$-Si substrate with local back-gate islands and then baked at 50 and 70 °C for 10 min each to remove moisture and residual PMMA, ensuring a pristine interface.

### Fabrication of monolayer $MoS_2$ memtransistors

To define the channel regions for the $MoS_2$ memtransistors, the substrate was spin-coated with PMMA and baked at 180 °C for 90 s. The resist was then exposed to an electron beam (e-beam) and developed using a 1:1 mixture of 4-methyl-2-pentanone (MIBK) and 2-propanol (IPA). The monolayer $MoS_2$ film was subsequently etched using sulfur hexafluoride ($SF_6$) at 5 °C for 30 s. Next, the sample was rinsed in acetone and IPA to remove the e-beam resist. To define the source and drain contacts, the sample is then spin-coated with methyl methacrylate (MMA) followed by A3 PMMA. Then, using e-beam lithography source and drain contacts were patterned and developed by using a 1:1 mixture of MIBK and IPA for the 60 s. 40 nm of Nickel (Ni) and 30 nm of Gold (Au) were deposited using e-beam evaporation. Finally, a lift-off process was performed to remove the evaporated Ni/Au except the source/drain patterns by immersing the sample in acetone for 30 min followed by IPA for another 30 min. Each island contains one $MoS_2$ memtransistor to allow for individual gate control.

### Monolithic integration

To define the connections between the respective nodes of $MoS_2$ memtransistors for the fabrication of the gustatory circuits, the substrate was spin-coated with MMA and PMMA, followed by e-beam lithography, developing using a 1:1 mixture of MIBK and IPA, and e-beam evaporation of 60 nm Ni and 30 nm Au. Finally, the e-beam resist was rinsed away by the lift-off process using acetone and IPA.

### Raman and photoluminescence (PL) spectroscopy

Raman and PL spectroscopy of the $MoS_2$ film were performed on a Horiba LabRAM HR Evolution confocal Raman microscope with a 532 nm laser. The power was 34 mW filtered at 5% to 1.7 mW. The objective magnification was ×100 with a numerical aperture of 0.9, and the grating had a spacing of 1800 gr/mm for Raman and 300 gr/mm for PL.

### Electrical characterization

Electrical characterization of the fabricated devices was performed in a Lake Shore CRX-VF probe station under atmospheric conditions using a Keysight B1500A parameter analyzer.

## Data availability

The datasets generated during and/or analyzed during the current study are available from the corresponding author on reasonable request.

## Code availability

The codes used for plotting the data are available from the corresponding authors on reasonable request.

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

## Acknowledgements

The work was supported by the Army Research Office (ARO) through Contract Number W911NF1810268 and National Science Foundation (NSF) through CAREER Award under Grant Number ECCS-2042154.

## Author contributions

S.D. conceived the idea and designed the experiments. S.G., A.P., D.S., A.W., and H.R. performed the measurements, analyzed the data, discussed the results, and agreed on their implications. All authors contributed to the preparation of the manuscript.

## Competing interests

The authors declare no competing interests.
