## [Peer Review File · Nature Communications]

REVIEWER COMMENTS

Reviewer #1 (Remarks to the Author):

The manuscript titled "An All 2D Bio-inspired Gustatory Circuit for Mimicking Physiology and Psychology of Feeding Behavior" written by Andrew Pannone et al. introduces the design of "electronic tongue" by using graphene-based chemical transistor as artificial taste receptor neuron, and the design of "electronic gustatory cortex" by single layer MoS₂ mem transistor, including physiologically driven "hunger neuron", psychologically driven "appetite neuron" and "feeding circuit". At the same time, the adaptive feed behavior is demonstrated by utilizing the analog and non-volatile programming capabilities of the MoS₂ mem transistor. The entire manuscript is clear in logic, complete in structure, and detailed in content. It is a very useful reference for researchers who are in contact with integrating transistors. Therefore, I recommend that this manuscript be accepted for publication after some revisions. Here are some comments.

- 1) In this work, the graphene device will inevitably produce some defects during the preparation process, thus affecting its initial properties (as shown in Figure 2c). So, do defects affect their sensing characteristics?
- 2) Please elaborate on the taste perception mechanism of graphene-based devices.
- 3) For the solution to be tested, will solutes precipitate on the graphene surface after water evaporation? Does the precipitate affect subsequent testing? What is the cycle life of the device? Please state and supplement the relevant tests in the manuscript.
- 4) Some original work on artificial nerves and recent progress on neuromorphic gustatory system needs to be discussed, e.g. Science 360(6392):998-1003; Nano Lett. 2023, 23, 1, 8–16.

Reviewer #2 (Remarks to the Author):

The author fabricates a complex of integration circuit using MoS₂ and Graphene to mimic the biological gustatory systems. Specifically, liquid gate Graphene chemitransistor is used as an electronic tongue for detecting food while memtransistor MoS₂ is used for memory and computing. Although the demonstration of this unique concept has not been reported previously, the practical application of this AI model is very uncertain.

1. The introduction poses whether an AI proficiency in physiology and psychology can bridge the gap between humans and machines. The usefulness of such an AI with emotional intelligence is questionable, and it remains to be seen if it would actually benefit humanity or potentially cause more problems for humans. The author also did not provide any solid references for this paradigm before stating further discussion.

2. The author provides the example of feeding behavior, which is influenced by multiple factors, as an illustration of psychology. Simply confining a logical equation like $F = H + HA$ to describe human feeding behavior is inadequate and absurd, as food preferences such as sweetness may not necessarily trigger feeding if they lead to negative health consequences like obesity and heart disease. It is uncertain whether the logical equation mentioned in this scenario can be validated without prior research in the fields of psychology and biology. Given that the practical benefits of emotional AI are still indefinite, the author's assertion that the research could be extended to other sensory systems such as visual, audio, and tactile is questionable.

3. Figure 2 depicts the author's use of liquids like coffee, lemon juice, and soy milk to represent sweet, bitter, and sour tastes in order to measure drain current on a Graphene-based gustatory taste device. While this method is commonly used for electronic tongue characterization, the manuscript does not thoroughly analyze the interaction between the chemical or organic molecules and Graphene in each liquid. As a result, the lack of quantitative analysis of graphene and molecules in the food liquids raises questions about the unclear mechanism of VDirac shift in graphene transistor.

4. As an illustration, soy milk contains numerous organic molecules and it is unclear which specific molecules are interacting with Graphene to cause the shift in $V_{\text{threshold}}$. And, other bitter, sweet, salty, and sour foods are possible to have relatively similar $V_{\text{threshold}}$ change as shown in this manuscript.

5. If food is being used as the gate liquid for a graphene transistor, the manuscript should include an explanation of the electrostatic mechanism in the liquid. For instance, it should address whether the leakage current is comparable to the drain current in this transistor concept. The V_c level for sour and Umami in Figure 2j is not clearly classified.

Reviewer #3 (Remarks to the Author):

In 'An all 2D bio-inspired gustatory circuit for mimicking physiology and psychology of feeding behavior', Pannone et al propose and experimentally demonstrate a hardware system that combines physiological response and psychological response. This is implemented by using two graphene chemitransistors, one mimics physiological signal (hunger) receptor, and the other mimics psychological signal (appetite) receptor. The two receptor outputs are fed into two MoS₂ data processing circuits that output zero or one based on predetermined tunable thresholds. The binary outputs from the processing circuits are further processed by a MoS₂ logic circuits for final decision making. The proposed idea of combining physiological response and psychological response in a single hardware is interesting. The presented 2D materials-based devices and circuits are of high quality. However, there are four major criticisms: the rationality of completely separating hunger and appetite, the scalability of the proposed method to more tastes, the rationality of determining hunger and appetite only by external stimuli.

First, the binary separation of hunger and appetite does not look reasonable. Appetite is the desire to consume food, which is strongly related to Hunger. But in the proposed method, appetite and hunger are completely separated.

Second, the scalability of the proposed method towards more tastes is unclear. Appetite is oversimplified to binary operation. $V_{AN} = 0.2V$ is chosen such that sweet is mostly above the appetite threshold while bitter is always below the appetite threshold. Using this binary coding scheme, it is unclear how to scale from two tastes to five tastes.

Third, hunger and appetite are determined by external stimuli. However, intuitively, hunger and appetite should be the intrinsic states of the sensory system. This criticism is less pronounced in the case of appetite, because the authors show the adaptive feeding behavior by tuning the threshold voltages (V_{TH-C}). But this criticism is severe in the case of defining hunger. In the proposed method, the system 'feels' hunger as long as aqueous food are presented. The system is full when all food is consumed, i.e. water evaporated. Once after that, if one more drop of liquid is added, the system feels hungry again. This is not mimicking the biological system. And the reason behind this drawback is that 'hunger' is solely determined by the presence of food, rather than determined by the intrinsic state of the sensory system.

Below are some minor comments:

1. $F = H + H_{bar} * A$ looks oversimplified. Is there any literature that support this?
2. How is electrical bias added to the liquid?
3. 'The unusual shape of the response curve is a direct consequence of the device-to-device variation'. Is this variation controllable? Does this variation imply calibration should be implemented for each graphene chemitransistor to find the feasible operation gate voltage? The VLGs are empirically chosen after measurement. And as the authors point out, the transfer curve of graphene chemitransistor is sensitive to variation of V_{Dirac} . This suggests that for every circuit, lots of measurements should be done before VLGs and threshold voltages (V_{TH-C}) can be properly chosen, making this method impractical.

4. What are the concentrations of diluted aqueous solutions?

5. 'Vc generated by graphene receptor are unique for each taste'. This might not validate the use of graphene chemitransistor as the electronic nose, as Vc also depends on concentration.

Response to Reviewers

Reviewer 1:

Comments: The manuscript titled “An All 2D Bio-inspired Gustatory Circuit for Mimicking Physiology and Psychology of Feeding Behavior” written by Andrew Pannone et al. introduces the design of "electronic tongue" by using graphene-based chemical transistor as artificial taste receptor neuron, and the design of " electronic gustatory cortex " by single layer MoS₂ mem transistor, including physiologically driven "hunger neuron", psychologically driven "appetite neuron" and "feeding circuit". At the same time, the adaptive feed behavior is demonstrated by utilizing the analog and non-volatile programming capabilities of the MoS₂ mem transistor. The entire manuscript is clear in logic, complete in structure, and detailed in content. It is a very useful reference for researchers who are in contact with integrating transistors. Therefore, I recommend that this manuscript be accepted for publication after some revisions. Here are some comments.

We would like to express our appreciation to the reviewer for the valuable feedback on the manuscript. The reviewer commended the manuscript for its clear logic, comprehensive structure, and detailed content. They have also recognized its significance as a valuable reference for researchers involved in integrating transistors. We sincerely thank the reviewer for the insightful comments and support.

1: In this work, the graphene device will inevitably produce some defects during the preparation process, thus affecting its initial properties (as shown in Figure 2c). So, do defects affect their sensing characteristics?

The reviewer is correct that the initial properties of a graphene chemitransistor device will be impacted by defects that are introduced throughout the fabrication process. The observed differences in the initial transfer characteristics of the two chemitransistors depicted in **Fig. 2c** can be attributed to contaminants that are present during wet transfer and fabrication procedures such as metal impurity dopants and polymer residues. Additionally, the wet transfer process may introduce nonidealities such as film strain or film wrinkles. **Fig. 2e-f** show the transfer characteristics of graphene chemitransistors 1 and 2, respectively, for all five taste categories. It is evident that the variations in initial electrical properties will cause variations in the electrical response to solutions that represent different tastes. While the response of two graphene

chemitranstistors to the same liquid gating solution will not be identical, it is important to note that a clear trend in the response of both chemitranstistors can be observed. This trend can be characterized quantitatively by evaluating key chemitranstistor figures of merit such as V_{Dirac} , as depicted in **Fig. 2h-i**. These variations are key towards obtaining nonmonotonic artificial taste receptor response curves that are crucial for our experimental demonstrations throughout the manuscript. The variability of device performance metrics across a population of ~200 graphene transtistors has been explored in one of our previous works [1].

2) Please elaborate on the taste perception mechanism of graphene-based devices.

We appreciate the reviewer's interest in the taste perception mechanism of graphene-based devices. In our study, we employ a liquid-gating technique to investigate the response characteristics of graphene chemitranstistors to different taste stimulants. Liquid-gating has been widely used in the study of field-effect transtistors (FETs) based on various nanomaterials [3, 4]. It offers several advantages, including the ability to achieve ultra-scaled effective oxide thickness (EOT) without encountering leakage problems associated with physically thin gate dielectrics. This is made possible by the formation of an electric double layer (EDL) at the solid/liquid interface. The EDL consists of electrons/holes in the FET channel and ions in the liquid media, separated by one or a few layers of solvent molecules that adhere to the channel surface and act as a gate dielectric. It should be noted that variations in liquid compositions can affect the EDL formation and lead to variations in the transfer characteristics, thus influencing the observed response to various taste stimulants. Moreover, it can also be stated that the change in the EDL formation may also modulate the EOTs for different chemical species, which directly affects the transconductances for the electron and hole branches in the ambipolar transfer characteristics of the graphene chemitranstistors. Furthermore, it is possible for charge transfer to occur between the graphene channel and various taste stimulant species. This phenomenon can result in different types and degrees of n- and p-type doping, consequently altering the Dirac-voltage point within a graphene chemitranstistor. **Fig. R1a-c**, respectively, show the electron and hole branch transconductances, $g_{m,n}$ and $g_{m,p}$, and Dirac-voltage, V_{Dirac} corresponding to different taste stimulants confirming that each taste stimuli has a distinct interaction with the graphene channel. This approach allows us to utilize graphene-based devices as constituent elements of an electronic tongue that is capable

of sensing and encoding taste information. We hope this clarifies the taste perception mechanism and the significance of liquid-gating in our study.

Figure R1. The peak transconductance (g_m) values were determined for the representative graphene chemitransistor in two cases: **a**) for the electron branch and **b**) for the hole branch in case of five different taste stimulants. These results confirm that each taste stimulus has a distinct effect. The nonoverlapping distributions of g_m in the graphene chemitransistor are caused by the interaction between the chemical species and the graphene channel. This interaction introduces surface scattering, which in turn affects the g_m . The specific g_m values obtained from the artificial taste receptor validate successful taste differentiation and support the utilization of graphene chemitransistors as an "electronic tongue". Additionally, the **c**) Dirac-voltage (V_{Dirac}) values have also been extracted for five taste stimulants, showing the nonoverlapping distributions of the voltages.

3) For the solution to be tested, will solutes precipitate on the graphene surface after water evaporation? Does the precipitate affect subsequent testing? What is the cycle life of the device? Please state and supplement the relevant tests in the manuscript.

We sincerely appreciate the valuable comments and inquiries raised by the reviewer. Regarding the concern about the solute precipitation on the graphene surface after water evaporation, we would like to emphasize to the reviewer that our testing protocol ensures comprehensive cleansing of the sample after each test run. The sample undergoes a rigorous washing process, including a 5-minute rinse with deionized water (DI), followed by a 5-minute treatment with acetone and a 3-minute treatment with isopropyl alcohol (IPA). This meticulous procedure guarantees the removal of any possible precipitates from the sample.

We have conducted a reusability measurement of the graphene chemitransistors for four different solution species for four cycles as shown in **Fig. R2**. A single cycle constitutes measuring the transfer characteristics using four different solution species. The electrical characterizations were

Figure R2. The reusability of graphene chemitransistor is shown for four different species for four cycles. A single cycle constitutes measuring the transfer characteristics at $V_{DS} = 10$ mV using four different species. Between each measurement, the device was washed using water, acetone, and isopropyl alcohol. Experiments were performed for a total of four cycles, showing the device characteristics for any given species remains mostly unaltered between the cycles.

performed for a total of four cycles, showing the chemitransistor characteristics for any given species remain mostly unaltered between the cycles. This indicates that the solutions used in the sensing process do not generate any defects that could potentially impact the sensing properties.

Fig. R3 shows optical images of the sample taken before and after the experiment as well as after cleaning the sample. Clearly, the washing procedure successfully removes any remaining precipitates, ensuring that subsequent testing remains unaffected. Finally, the life cycle of the representative chemitransistor was evaluated through an endurance measurement, where a chemitransistor was measured for 100 cycles with sugar solution as shown in **Fig. R4**.

We have included the above discussions and results in the revised manuscript and **Supplementary Information 3-5**.

Figure R3. Optical images capturing the sample throughout the experiment. **a)** Initial state of the sample before the experiment. **b)** Optical image of the sample showing precipitation of the solution. **c)** Optical image of the sample after thorough wash with DI water, acetone, and IPA.

Figure R4. Endurance plot of 100 cycles showing consistent transfer characteristics of the graphene chemitristor for sugar solution at $V_{DS} = 500$ mV.

4) Some original work on artificial nerves and recent progress on neuromorphic gustatory system needs to be discussed, e.g., *Science* 360(6392):998-1003; *Nano Lett.* 2023, 23, 1, 8–16.

We sincerely appreciate the valuable input provided by the reviewer. The recommendation to include discussion regarding original work on artificial nerves and recent progress in the field of neuromorphic gustatory systems, such as *Science* 360(6392):998-1003 and *Nano Lett.* 2023, 23, 1, 8–16, has been incorporated into our introduction section. We acknowledge the significance of these exemplary works in enhancing the quality and depth of our research. Their contributions have greatly enriched our understanding and provided valuable insights into the advancements in artificial nerves and the emerging field of neuromorphic gustatory systems.

We have cited these articles in our revised manuscript.

Reviewer 2:

Comments: The author fabricates a complex integration circuit using MoS₂ and Graphene to mimic the biological gustatory systems. Specifically, liquid gate Graphene chemitransistor is used as an electronic tongue for detecting food while memtransistor MoS₂ is used for memory and computing. Although the demonstration of this unique concept has not been reported previously, the practical application of this AI model is very uncertain.

We sincerely appreciate the reviewer's recognition of our work in fabricating a complex integrated circuit using MoS₂ and Graphene to mimic biological gustatory systems. We acknowledge their observation that the demonstration of this unique concept has not been previously reported. This highlights the novelty and originality of our research, and we are honored by their acknowledgment of our contribution to the field. We acknowledge that further investigation and exploration are required to fully understand the practical applications of this AI model. We value the reviewer's feedback and will continue to pursue further advancements and potential applications of our work.

1. The introduction poses whether an AI proficiency in physiology and psychology can bridge the gap between humans and machines. The usefulness of such an AI with emotional intelligence is questionable, and it remains to be seen if it would benefit humanity or potentially cause more problems for humans. The author also did not provide any solid references for this paradigm before stating further discussion.

The reviewer posed an interesting question. The concept of machines possessing emotional intelligence is an area of ongoing research and development [5, 6]. While machines cannot experience emotions in the same way humans do, incorporating emotional intelligence into their capabilities can offer several potential benefits especially with growing dominance of artificial intelligence (AI) in all spheres of our lives. Machines with emotional intelligence can understand and respond to human emotions, making interactions more natural and meaningful. Emotional intelligence also enables machines to understand individual needs and preferences. By recognizing emotions, they can provide tailored recommendations, suggestions, or support. For instance, a virtual assistant with emotional intelligence can detect if a person is stressed and recommend food that can uplift their mood [7]. Likewise, emotional intelligence can be applied in various ways within the food industry to enhance customer experiences and drive business success [8]. By

incorporating emotional intelligence into the menu development process, restaurants can create dishes and experiences that evoke positive emotions, enhancing customer satisfaction and loyalty [9]. Emotional intelligence can also accelerate culinary innovation by understanding the emotional impact of different flavors, textures, and food combinations. While the benefits of machines having emotional intelligence are promising, we agree with the reviewer that it is essential to consider ethical considerations, data privacy, and the potential impact on human employment and relationships. Striking a balance between the benefits of emotional intelligence and responsible implementation is crucial for creating a positive and sustainable future with intelligent machines. We apologize for not including adequate references before discussing the emotional intelligence paradigm.

We have included some of the above discussion and references in the revised manuscript and *Supplementary Information 13*.

2. The author provides the example of feeding behavior, which is influenced by multiple factors, as an illustration of psychology. Simply confining a logical equation like $F = H + HA$ to describe human feeding behavior is inadequate and absurd, as food preferences such as sweetness may not necessarily trigger feeding if they lead to negative health consequences like obesity and heart disease. It is uncertain whether the logical equation mentioned in this scenario can be validated without prior research in the fields of psychology and biology. Given that the practical benefits of emotional AI are still indefinite, the author's assertion that the research could be extended to other sensory systems such as visual, audio, and tactile is questionable.

We acknowledge the reviewer's concern and agree that feeding is a complex behavior influenced by various factors. Therefore, reducing it to a simple logical equation does not provide a complete depiction of the feeding process. However, the primary objective of our work is to introduce the concept of emotional intelligence and demonstrate the integration of psychology and physiology in decision making, while maintaining readability and avoiding overwhelming complexity. It is important to note that future research in the fields of psychology and biology can potentially incorporate multiple state variables and different logic circuits to provide a more comprehensive understanding of the feeding process. In our study, we relied on existing research on feeding [10-14] to develop a higher-level abstraction of this process.

Moreover, as we have discussed in response to previous comments, there are practical benefits associated with emotional AI, but responsible conduct is crucial to ensure their positive utilization. We believe that extending this research to other sensory systems, such as visual, audio, and tactile, represents the future of AI. However, we are also aware of the ethical and moral considerations that must be taken into account in this advancement.

3. Fig. 2 depicts the author's use of liquids like coffee, lemon juice, and soy milk to represent sweet, bitter, and sour tastes in order to measure drain current on a Graphene-based gustatory taste device. While this method is commonly used for electronic tongue characterization, the manuscript does not thoroughly analyze the interaction between the chemical or organic molecules and Graphene in each liquid. As a result, the lack of quantitative analysis of graphene and molecules in the food liquids raises questions about the unclear mechanism of V_{Dirac} shift in graphene transistor.

Thank you for raising the concern regarding the interaction between the chemical or organic molecules and graphene in the liquids used in our gustatory taste device. We appreciate your observation and would like to address your question.

In our study, we employ a liquid-gating technique to investigate the response characteristics of graphene chemitransistors to different taste stimulants. Liquid-gating has been widely used in the study of field-effect transistors (FETs) based on various nanomaterials [3, 4]. It offers several advantages, including the ability to achieve ultra-scaled effective oxide thickness (EOT) without encountering leakage problems associated with physically thin gate dielectrics. This is made possible by the formation of an electric double layer (EDL) at the solid/liquid interface. The EDL consists of electrons/holes in the FET channel and ions in the liquid media, separated by one or a few layers of solvent molecules that adhere to the channel surface and act as a gate dielectric. It should be noted that variations in liquid compositions can affect the EDL formation and lead to variations in the transfer characteristics, thus influencing the observed response to various taste stimulants. Moreover, it can also be stated that the change in the EDL formation may also modulate the EOTs for different chemical species, which directly affects the transconductances for the electron and hole branches in the ambipolar transfer characteristics of the graphene chemitransistors. Furthermore, it is possible for charge transfer to occur between the graphene channel and various taste stimulant species. This phenomenon can result in different types and

Figure R5. The peak transconductance (g_m) values were determined for the representative graphene chemitransistor in two cases: **a)** for the electron branch and **b)** for the hole branch in case of five different taste stimulants. These results confirm that each taste stimulus has a distinct effect. The nonoverlapping distributions of g_m in the graphene chemitransistor are caused by the interaction between the chemical species and the graphene channel. This interaction introduces surface scattering, which in turn affects the g_m . The specific g_m values obtained from the artificial taste receptor validate successful taste differentiation and support the utilization of graphene chemitransistors as an "electronic tongue". Additionally, the **c)** Dirac-voltage (V_{Dirac}) values have also been extracted for five taste stimulants, showing the nonoverlapping distributions of the voltages.

degrees of n- and p-type doping, consequently altering the Dirac-voltage point within a graphene chemitransistor. **Fig. R5a-c**, respectively, show the electron and hole branch transconductances, $g_{m,n}$ and $g_{m,p}$, and Dirac-voltage, V_{Dirac} corresponding to different taste stimulants confirming that each taste stimuli has a distinct interaction with the graphene channel. This approach allows us to utilize graphene-based devices as constituent elements of an electronic tongue that is capable of sensing and encoding taste information. We hope this clarifies the taste perception mechanism and the significance of liquid-gating in our study.

4. As an illustration, soy milk contains numerous organic molecules, and it is unclear which specific molecules are interacting with Graphene to cause the shift in $V_{\text{Threshold}}$. And other bitter, sweet, salty, and sour foods are possible to have relatively similar $V_{\text{Threshold}}$ change as shown in this manuscript.

Thank you for your comment regarding the specific organic molecules in soy milk that may be interacting with graphene to cause the shift in $V_{\text{Threshold}}$. We appreciate your observation and would like to address this point.

However, we would like to clarify a minor error in the statement. In our study, it is not soy milk but rather soy sauce that we have utilized as one of the taste stimuli representing umami taste.

Indeed, identifying the specific molecules in soy sauce that interact with graphene to induce the V_{Dirac} shift is a complex task. While we acknowledge the importance of understanding precise molecular interactions, it is challenging to isolate and identify each individual molecule's contribution within a complex mixture such as soy sauce. However, it is important to note that the V_{Dirac} shift can vary for different solution species due to several factors. Each solution, including soy sauce, has its unique pH value, and graphene's exceptional pH sensing capabilities come into play [15-17]. Additionally, the composition of each solution comprises different chemical molecules that interact distinctively with graphene, leading to the formation of an electric double layer (EDL). The ultra-scaled effective oxide thickness (EOT) arises from this EDL at the solid/liquid interface, where the EDL consists of electrons/holes in the FET channel and ions in the liquid media. These layers are typically separated by one or a few layers of solvent molecules adhering to the channel surface, acting as a gate dielectric. As a result, the capacitance values of the EDL closely resemble those of ultra-thin and high-k solid-state gate dielectrics, enabling precise electrostatic control in the chemitransistor channel. It is important to mention that not only the V_{Dirac} shift but also the transconductance (g_m) values vary with each distinct solution. Therefore, a comprehensive analysis of the transconductance values corresponding to different solution species would provide further insights into the mechanism underlying the V_{Dirac} shift in graphene chemitransistors. While we strive to deepen our understanding of the specific molecular interactions, the complexity of food liquids, such as soy sauce, makes it challenging to isolate individual components and their contributions to the V_{Dirac} shift.

5. If food is being used as the gate liquid for a graphene transistor, the manuscript should include an explanation of the electrostatic mechanism in the liquid. For instance, it should address whether the leakage current is comparable to the drain current in this transistor concept. The V_C level for sour and Umami in Figure 2j is not clearly classified.

Thank you for your valuable feedback regarding the electrostatic mechanism in the liquid and the classification of V_C levels for sour and umami in **Fig. 2j** of our manuscript. We appreciate your comments and would like to address them.

In our graphene FET devices, the combination of higher carrier mobility, single-layer thickness, impermeability to liquids, and chemical stability contributes to a lower gate leakage current compared to drain current as shown in **Fig. R6**.

We have also included these figures in *Supplementary Information 2*.

Figure R6. Comparison of gate leakage current for five different tastes and DI water. Leakage current was found to be negligible on the order of a few nA.

Regarding **Fig. 2j**, the reviewer is correct that at the specific V_{LG} utilized in the manuscript, it is difficult to differentiate sour and umami. The V_C levels for sour and Umami are indeed 0.18V and 0.16V, respectively, at V_{LG} (liquid gate voltage) of 0.1V. However, the input to the artificial gustatory taste receptor can be modulated to accentuate different taste response characteristics. **Fig. 2g** shows the output characteristics of the artificial gustatory taste receptor when V_{LG} is varied in the range of -0.5 V to 0.5 V. Clearly, several regions are present that can be chosen to highlight differentiability in sour and umami tastes. For example, V_C is 0.275V and 0.285V at a V_{LG} of -0.25V for sour and umami tastes, respectively. The specific choice of V_{LG} is crucial to demonstrate differentiation between different tastes. We apologize for any confusion caused and have provided additional discussion to clarify this concept in the revised manuscript.

Reviewer 3:

Comments: In 'An all 2D bio-inspired gustatory circuit for mimicking physiology and psychology of feeding behavior', Pannone et al propose and experimentally demonstrate a hardware system that combines physiological response and psychological response. This is implemented by using two graphene chemitransistors, one mimics physiological signal (hunger) receptor, and the other mimics psychological signal (appetite) receptor. The two receptor outputs are fed into two MoS2 data processing circuits that output zero or one based on predetermined tunable thresholds. The binary outputs from the processing circuits are further processed by a MoS2 logic circuit for final decision making. The proposed idea of combining physiological response and psychological response in a single hardware is interesting. The presented 2D materials-based devices and circuits are of high quality. However, there are four major criticisms: the rationality of completely separating hunger and appetite, the scalability of the proposed method to more tastes, the rationality of determining hunger and appetite only by external stimuli. First, the binary separation of hunger and appetite does not look reasonable. Appetite is the desire to consume food, which is strongly related to Hunger. But in the proposed method, appetite and hunger are completely separated.

Second, the scalability of the proposed method towards more tastes is unclear. Appetite is oversimplified to binary operation. $V_{AN} = 0.2V$ is chosen such that sweet is mostly above the appetite threshold while bitter is always below the appetite threshold. Using this binary coding scheme, it is unclear how to scale from two tastes to five tastes.

Third, hunger and appetite are determined by external stimuli. However, intuitively, hunger and appetite should be the intrinsic states of the sensory system. This criticism is less pronounced in the case of appetite because the authors show the adaptive feeding behavior by tuning the threshold voltages (V_{TH-C}). But this criticism is severe in the case of defining hunger. In the proposed method, the system 'feels' hunger as long as aqueous food are presented. The system is full when all food is consumed, i.e. water evaporated. Once after that, if one more drop of liquid is added, the system feels hungry again. This is not mimicking the biological system. And the reason behind this drawback is that 'hunger' is solely determined by the presence of food, rather than determined by the intrinsic state of the sensory system.

We genuinely appreciate the reviewer's insightful feedback on our work. We are grateful for the recognition of the work as high quality and interesting. We value the critical observations regarding the separation of hunger and appetite, scalability to more tastes, and the determination of hunger and appetite solely based on external stimuli. We have addressed these aspects below.

The rationality of completely separating hunger and appetite: Our rationale for separating hunger and appetite is as follows: Hunger is a physiological sensation that arises from the body's need for nourishment. It is primarily driven by biological factors and the body's energy requirements. When you experience hunger, it is a signal from your body that it needs food to meet its energy needs and maintain proper function. Hunger is often accompanied by physical sensations like stomach growling, feeling lightheaded, or having low energy levels. Appetite, on the other hand, refers to the desire or preference for specific types of food or the urge to eat. Unlike hunger, which is primarily driven by biological factors, appetite is influenced by a combination of physiological, psychological, and social factors. Appetite is shaped by factors such as sensory cues (smell, taste, and appearance of food), learned preferences, emotional states, cultural influences, and environmental cues. Appetite can vary greatly from person to person and may not always align with actual physiological hunger. In summary, hunger is the physiological sensation of needing food to satisfy energy requirements, while appetite refers to the desire or preference for specific foods or the urge to eat, influenced by various factors beyond just physiological need. We agree that it is not possible to separate hunger and appetite. In fact, the same is true for physiology and psychology and hence intellectual and emotional intelligence. As the physical connection between hunger and appetite becomes clearer from studies in psychology and biology, it will be possible for us to refine the expression for feeding in our future studies. However, considering the manuscript's primary focus on developing hardware components to capture different aspects of intelligence, we made the decision to treat hunger and appetite as distinct concepts. This approach was chosen to ensure readability and to prevent excessive complexity. By keeping hunger and appetite separate within the context of the manuscript, we aimed to provide a clearer and more concise presentation of the research.

We have included the above discussion in *Supplementary Information 14*.

The scalability of the proposed method to more tastes: Our investigation has examined how a graphene chemitransistor responds to a diverse range of taste stimulants, aiming to emphasize the

Figure R7. Transfer characteristics of graphene chemitransistors for **a)** four different sugar solutions and **b)** umami solutions at $V_{DS} = 10\text{ mV}$.

extensive capabilities of graphene as a versatile electronic sensor. It has the ability to detect and distinguish between different tastes, thereby functioning as a “universal electronic tongue”. Alongside the five taste stimulants (salt, sweet, sour, bitter, and umami) described in the manuscript, we have also evaluated the chemitransistor's performance with four different sugar solutions as well as four distinct milk (umami) solutions. To analyze the chemitransistor's behavior, we plot the transfer characteristics of a representative device against the liquid top gate voltage (V_{LG}) for all the afore mentioned taste stimulants as shown in **Fig. R7a-b**. It is worth noting that the initial results demonstrate the ability of our electronic tongue, utilizing graphene chemitransistors, to effectively differentiate and detect various tastes through the analysis of transfer characteristics. However, a more detailed and comprehensive analysis of the data and interpretation is planned as a future study. This will involve examining the specific response patterns, thresholds, and sensitivity of the chemitransistor to each taste, further enhancing our understanding of the electronic tongue's capabilities.

We have included the above figures in **Supplementary Information 11**.

The rationality of determining hunger and appetite only by external stimuli: We agree that hunger is primarily driven by biological factors and the body's internal energy requirements, whereas appetite can be driven by a combination of physiological, psychological, and social factors including sensory cues (smell, taste, and appearance of food), learned preferences, emotional states, cultural influences, and environmental cues. So, it is not fair to treat both as external stimuli. We acknowledge that stimulating hunger in artificial systems can be challenging, as hunger is a

complex physiological process. However, there are a few approaches that can be considered. For example, artificial systems can simulate hunger by monitoring and responding to specific physiological parameters associated with hunger. For example, tracking blood glucose levels, ghrelin (hunger hormone) levels, or even neural activity related to hunger signals can provide input for stimulating hunger in the system. However, considering the manuscript's primary focus on developing hardware components to capture different aspects of intelligence, we made the decision to treat hunger and appetite as external stimuli so that we can perform the proof-of-concepts experiments.

We have included the above discussion in *Supplementary Information 14*.

1. $F = H + \bar{H} * A$ looks oversimplified. Is there any literature that supports this?

We sincerely appreciate the reviewer's comment regarding the oversimplification of the feeding equation $F = H + \bar{H} * A$. We admit that feeding is a complex behavior influenced by various factors. Therefore, reducing it to a simple logical equation does not provide a complete depiction of the feeding process. However, the primary objective of our work is to introduce the concept of emotional intelligence and demonstrate the integration of psychology and physiology in decision making, while maintaining readability and avoiding overwhelming complexity. It is important to note that future research in the fields of psychology and biology can potentially incorporate multiple state variables and different logic circuits to provide a more comprehensive understanding of the feeding process. In our study, we relied on existing research on feeding [10-14] to develop a higher-level abstraction of this process. While the direct causal relationship expressed in the equation may appear simplified, it is worth noting that constructing logic diagrams based on observations can provide insights into the interplay between psychological factors and sensory pathways, ultimately influencing decision making. Therefore, our work aims to contribute to the understanding of these intricate connections, despite limited literature supporting this hypothesis. We sincerely appreciate the reviewer's thoughtful comments, as they encourage further exploration and refinement in this fascinating area of research.

2. How is electrical bias added to the liquid?

We appreciate the opportunity to clarify this aspect of our experimental setup, and we thank the reviewer for raising this question, as it allows us to provide additional details regarding the methodology employed in our research. **Fig. R8a** illustrates the cross-sectional schematic of a graphene chemitransistor. **Fig. R8b** shows an optical image of a chemitransistor array during a liquid gating experiment with source, drain, and liquid gate electrodes labelled. It is important to note that in **Fig R8b**, the chemical solution is in contact with the common gate electrode that extends to the outer periphery for ease of access where a probe provides the necessary electrical bias. Moreover, to mitigate the impact of liquid media on gate-leakage current, an alumina capping layer has been deposited. This layer serves as an insulator, effectively isolating the gate electrode from the source/drain electrodes. By introducing this protective barrier, the risk of undesirable current leakage between these components is substantially minimized. As a result, the system achieves enhanced performance and greater reliability, ensuring optimal functionality even in the presence of liquid media.

In our work, we utilized a representative liquid media, NaCl solution for salt, Coffee for bitter, sugar for sweet, soy sauce for umami and lemon juice for sour, to gate the graphene chemitransistor. To introduce the electrical bias, a droplet of the liquid solution was carefully placed on the chemitransistor using a micropipette. The transfer characteristics of the representative chemitransistors were measured by considering the source-to-drain current (I_{DS}) as a function of the gate voltage (V_{TG}) applied to the liquid media, while maintaining a constant source-to-drain bias (V_{DS}) of 500 mV. As a result of the applied bias to the liquid gate, an EDL forms at the graphene/chemical interface, which consists of electrons/holes in the FET channel and ions in the liquid media, separated by a few layers of solvent molecules adhering to the channel surface and acting as the gate dielectric.

Figure R8. a) Device schematic and b) optical image of the graphene chemitransistor.

3. *'The unusual shape of the response curve is a direct consequence of the device-to-device variation'. Is this variation controllable? Does this variation imply calibration should be implemented for each graphene chemitransistor to find the feasible operation gate voltage? The V_{LGs} are empirically chosen after measurement. And as the authors point out, the transfer curve of graphene chemitransistor is sensitive to variation of V_{Dirac} . This suggests that for every circuit, lots of measurements should be done before V_{LGs} and threshold voltages (V_{TH-C}) can be properly chosen, making this method impractical.*

We appreciate your comment regarding the device-to-device variation and its impact on the response curve of the graphene chemitransistor. We would like to clarify that while this variation does contribute to the unusual shape of the response curve, it is controllable to some extent, namely through the improvement in graphene synthesis, film transfer, and fabrication processes. In fact, in our previous work published in Nature Electronics, 1-11, 2021 [1], we have investigated and addressed the factors associated with device-to-device variation in graphene-based devices. While eliminating device-to-device variation would ensure consistent and reproducible characteristics of graphene chemitransistors, it would also eliminate the non-monotonic shape of the voltage divider curve, which we have utilized as an artificial taste receptor neuron. In this study, we specifically aimed to obtain varying responses from two graphene chemitransistors connected in series, leading to a changing V_C value. If both devices exhibited the same response, a constant V_C value would be obtained, which would not align with the experimental design of this study. In order to reintroduce slight variation between the two chemitransistors, we propose to exploit the memristive properties of graphene as discussed in our other work published in Nature Communications (2020) 11, 5474, where we have achieved over 16 memory states in graphene with adequate retention and endurance [18]. Therefore, while in the current study, we intentionally utilized direct device-to-device variation to achieve the desired monotonic shape of the voltage divider curve, our ultimate objective is to engineer the variation as explained above.

The liquid gate voltage range feasible for the operation of the artificial taste receptor was carefully selected to encapsulate the critical features of the response characteristics while minimizing the negative impacts of potential leakage currents at higher liquid gate voltages. **Fig. R9** shows the leakage current plots for all taste categories which show negligible values ($\sim 1-10$ nA) within the selected voltage range. The reviewer raises a valid concern that the liquid voltage range of interest may change on a device-to-device basis. Therefore, we have conducted an experiment to confirm

Figure R9. Comparison of gate leakage current for five different tastes and DI water, as a function of V_{LG} .

our choice of operating voltage range. **Fig. R10a** shows the transfer characteristics of 5 graphene chemitransistor devices measured in the range of -0.5V to 0.5V. Please note that V_{Dirac} values in addition to both the n - and p -branches for all 5 graphene chemitransistors are clearly visible which further affirms the selected voltage range. In addition, the leakage current values for all 5 chemitransistor devices, as shown in **Fig. R10b**, also remain negligible. Clearly, this range captures the of the suitable operating voltage range across a population of devices without requiring any calibration.

Figure R10. (a) Transfer characteristics of five graphene chemitransistors for milk solutions at $V_{DS} = 10$ mV. (b) gate leakage current as a function of V_{LG} for five graphene chemitransistors.

We have included the leakage current figures in **Supplementary Information 2**.

As far as the practical/real-time application of our proposed technology is concerned, we do acknowledge the impediments imposed by variations in V_{Dirac} and as a result our choice of V_{LG} . However, we would like to point out here that these variations in 2D-based devices are commonly reported in literature. In this context, it is therefore expected that future work would focus on optimizing and eliminating these variations which would surely enhance the feasibility of our proposed method.

4. What are the concentrations of diluted aqueous solutions?

The reviewer has raised a valid point here that requires further clarification. Note that in our study, we utilized a 10 mM concentration of NaCl solution. Soy sauce, lemon juice, coffee, and sugar representing umami, sour, bitter, and sweet tastes, respectively, were commercially purchased. Unfortunately, due to the inherent composition complexities of these commercially available products, it is challenging to calculate their exact concentration. For example, composition and concentrations of compounds in different types of coffee can vary based on the choice of bean and their roasting techniques. Similarly, sugar packets generally contain varying amounts of glucose, sucrose, and fructose, among other components. Therefore, determining exact concentration for each of these solutions becomes infeasible. While we understand that this information is certainly valuable, the aim of our study is to focus more on the qualitative aspects of taste perception rather than quantitative concentration analysis.

5. 'V_c generated by graphene receptor are unique for each taste'. This might not validate the use of graphene chemitransistor as the electronic nose, as V_c also depends on concentration.

We genuinely appreciate the question regarding the use of graphene chemitransistors as an electronic tongue and the potential influence of concentration on the generated V_C values. As proof of concept, we have evaluated the response of graphene-based taste receptors to different dilution of a salt solution as shown in **Fig. R11**. Clearly, the concentration of taste stimulants has a distinct influence on the initial transfer characteristics of the chemitransistor. Therefore, since both graphene chemitransistors in a voltage divider configuration will likely differ in their transfer characteristics as a result of the device-to-device variation, the observed V_C values for different

concentrations of taste stimulants are also expected to be unique, which in turn can impact the feeding behavior.

Figure R11. Transfer characteristics of graphene-based taste receptors to different concentrations (1mM, 10mM, 100mM) of salt solution within a gate voltage range of -0.5V to 0.5V at $V_{DS} = 10$ mV.

However, conducting a more extensive study on the concentration's impact on various taste stimulants is beyond the scope of the present manuscript. We plan to undertake this investigation as part of our future research on this topic. The main focus of this manuscript is to demonstrate the temporal changes in the characteristics plot as the liquid solution evaporates, using V_C as a proof of concept. Our objective was to highlight the distinct response patterns generated by the graphene receptor for different tastes, rather than providing a comprehensive analysis of concentration-dependent effects. We acknowledge the importance of considering concentration as a factor in electronic tongue applications and its potential influence on V_C values. Nevertheless, our current work aimed to establish the feasibility of graphene chemitransistors in detecting taste variations, laying the groundwork for future studies that can delve deeper into concentration-dependent effects.

We have included the figure in **Supplementary Information 12**.

References:

- [1] A. Dodda, S. Subbulakshmi Radhakrishnan, T. F. Schranghamer, D. Buzzell, P. Sengupta, and S. J. N. E. Das, "Graphene-based physically unclonable functions that are reconfigurable and resilient to machine learning attacks," vol. 4, no. 5, pp. 364-374, 2021.
- [2] J. Lieb *et al.*, "Ionic-Liquid Gating of InAs Nanowire-Based Field-Effect Transistors," vol. 29, no. 3, p. 1804378, 2019.
- [3] C.-S. Lee, S. K. Kim, and M. Kim, "Ion-sensitive field-effect transistor for biological sensing," *Sensors*, vol. 9, no. 9, pp. 7111-7131, 2009.
- [4] W. Fu *et al.*, "High mobility graphene ion-sensitive field-effect transistors by noncovalent functionalization," *Nanoscale*, vol. 5, no. 24, pp. 12104-12110, 2013.
- [5] B. A. Erol, A. Majumdar, P. Benavidez, P. Rad, K. K. R. Choo, and M. Jamshidi, "Toward Artificial Emotional Intelligence for Cooperative Social Human–Machine Interaction," *IEEE Transactions on Computational Social Systems*, vol. 7, no. 1, pp. 234-246, 2020, doi: 10.1109/TCSS.2019.2922593.
- [6] D. Schuller and B. W. Schuller, "The age of artificial emotional intelligence," *Computer*, vol. 51, no. 9, pp. 38-46, 2018.
- [7] E. Brynjolfsson and A. McAfee, "Artificial intelligence, for real," *Harvard business review*, vol. 1, pp. 1-31, 2017.
- [8] I. Kumar, J. Rawat, N. Mohd, and S. Husain, "Opportunities of artificial intelligence and machine learning in the food industry," *Journal of Food Quality*, vol. 2021, pp. 1-10, 2021.
- [9] K. Berezina, O. Ciftci, and C. Cobanoglu, "Robots, artificial intelligence, and service automation in restaurants," in *Robots, artificial intelligence, and service automation in travel, tourism and hospitality*: Emerald Publishing Limited, 2019.
- [10] O. Fu, Y. Minokoshi, and K.-i. Nakajima, "Recent advances in neural circuits for taste perception in hunger," *Frontiers in neural circuits*, vol. 15, p. 609824, 2021.
- [11] P. Masek and A. C. Keene, "Gustatory processing and taste memory in *Drosophila*," *Journal of neurogenetics*, vol. 30, no. 2, pp. 112-121, 2016.
- [12] L. Wang, H. Sato, Y. Satoh, M. Tomioka, H. Kunitomo, and Y. Iino, "A gustatory neural circuit of *Caenorhabditis elegans* generates memory-dependent behaviors in Na⁺ chemotaxis," *Journal of Neuroscience*, vol. 37, no. 8, pp. 2097-2111, 2017.
- [13] P. K. Shiu, G. R. Sterne, S. Engert, B. J. Dickson, and K. Scott, "Taste quality and hunger interactions in a feeding sensorimotor circuit," *Elife*, vol. 11, p. e79887, 2022.
- [14] A. J. Oliveira-Maia, C. D. Roberts, S. A. Simon, and M. A. Nicolelis, "Gustatory and reward brain circuits in the control of food intake," *Advances and technical standards in neurosurgery*, pp. 31-59, 2011.
- [15] P. K. Ang, W. Chen, A. T. S. Wee, and K. P. J. J. o. t. A. C. S. Loh, "Solution-gated epitaxial graphene as pH sensor," vol. 130, no. 44, pp. 14392-14393, 2008.
- [16] P. Salvo *et al.*, "Graphene-based devices for measuring pH," vol. 256, pp. 976-991, 2018.
- [17] N. Lei, P. Li, W. Xue, J. J. M. s. Xu, and technology, "Simple graphene chemiresistors as pH sensors: fabrication and characterization," vol. 22, no. 10, p. 107002, 2011.
- [18] T. F. Schranghamer, A. Oberoi, and S. J. N. c. Das, "Graphene memristive synapses for high precision neuromorphic computing," vol. 11, no. 1, p. 5474, 2020.

REVIEWERS' COMMENTS

Reviewer #1 (Remarks to the Author):

The authors have well revised the manuscript and answered the reviewers' comments. Thus, it can be accepted.

Reviewer #2 (Remarks to the Author):

The authors have carefully addressed the questions and revised the manuscript accordingly. In its present form, the manuscript is acceptable for publication.

Reviewer #3 (Remarks to the Author):

The authors have addressed all my questions. Although some questions are not completely addressed (e.g. hunger as external stimuli, scalability issue caused by binary operation and system complexity when concentration variation is considered), I agree with the authors that these are questions for future study, and the manuscript in its current simplified form serves as a valuable finding that points toward the direction of synergizing physiological and psychological signals for enhanced artificial intelligence capability. Therefore, I would recommend the publication of this manuscript.

Reviewer #1 (Remarks to the Author):

The authors have well revised the manuscript and answered the reviewers' comments. Thus, it can be accepted.

We express our gratitude to the reviewer for appreciating our response and recommending the acceptance of our manuscript.

Reviewer #2 (Remarks to the Author):

The authors have carefully addressed the questions and revised the manuscript accordingly. In its present form, the manuscript is acceptable for publication.

We express our gratitude to the reviewer for appreciating our response and recommending the acceptance of our manuscript.

Reviewer #3 (Remarks to the Author):

The authors have addressed all my questions. Although some questions are not completely addressed (e.g. hunger as external stimuli, scalability issue caused by binary operation and system complexity when concentration variation is considered), I agree with the authors that these are questions for future study, and the manuscript in its current simplified form serves as a valuable finding that points toward the direction of synergizing physiological and psychological signals for enhanced artificial intelligence capability. Therefore, I would recommend the publication of this manuscript.

We express our gratitude to the reviewer for appreciating our response and recommending the acceptance of our manuscript.